# Methanolic crude extract of *Litsea Monopetala* leaves combats oxidative stress, clot formation, inflammation and stool frequency in animal model

Sarder Arifuzzaman◉°*, Zubair Khalid Labu°, Banani Das Ani, Samira Karim, Md. Tarekur Rahman

Department of Pharmacy, World University of Bangladesh, Uttara, Dhaka, Bangladesh

° These authors contributed equally to this work.
* sarder.arifuzzaman@pharmacy.wub.edu.bd; arifpharmju@gmail.com

## Abstract

### Introduction

*Litsea monopetala* (LM) leaves are used in traditional medicine system in the South Asian region for treating ailments such as digestive issues, respiratory problems and skin disorders. In this study, we investigated the possible antioxidant, thrombolytic, analgesic and antidiarrheal properties of the methanolic extract of *LM* leaves.

### Method

We assessed the antioxidant activity using DPPH (2,2-diphenyl-1-picrylhydrazyl) free radical scavenging and total phenolic content tests, while thrombolytic activity was evaluated by clot lysis assays. The *in-vivo* analgesic and antidiarrheal activities were tested by two standard methods, e.g., acetic acid-induced and castor oil-induced animal model, respectively. Prior to *in vivo* and *in vitro* evaluation of the pharmacological activities phytochemical screening was also performed to estimate the bioactive compounds (e.g., phenol, carbohydrates, reducing sugars, tannins, alkaloids, flavonoids, saponins and steroids) present.

### Results

Among the tested phytochemicals, our results reveal carbohydrates, alkaloids, flavonoids and tannins as the major phytocompounds present in the extract. Within the different solvent extractives, the methanolic extract exhibited DPPH free radical scavenging features with an $IC_{50}$ of 8.99 µg/ml compared to ascorbic acid, an $IC_{50}$ of 13.38 µg/ml. At 500 mg/kg dose, the extract produced 67.05% decrease in the frequency of acetic acid-induced writhing while diclofenac sodium showed decrease by 74.25%. The extract also significantly ($P < 0.01$) decreased the frequency of castor oil-induced

**Data availability statement:** All relevant data are within the manuscript and its Supporting Information files.

**Funding:** The author(s) received no specific funding for this work.

**Competing interests:** The authors have declared that no competing interests exist.

diarrhea in compared to the standard drug of loperamide. Finally, the clot lysis assay with the methanolic extract demonstrated an increase in the thrombolytic activity by 40.79% compared to streptokinase, which increased by 69.52%.

## Conclusion

Overall, this study shows promise that the methanolic crude extract of *LM* leaves may contribute to the alternative or additive strategy to modulate conditions of oxidative stress, thrombolytic, inflammation and diarrhea. Further comprehensive investigation is necessary to clarify the exact mechanisms of action and the phytochemical composition of *LM* leaves.

---

## Introduction

*Litsea* species have been used worldwide in traditional medicine for the treatment of various diseases such as influenza, stomachaches, diarrhea, diabetes, vomiting, bone pain, inflammation, central nervous system disorders, and other conditions [1,2]. These species contain a multitude of biologically active substances, including butanolides found in the leaves of *Litsea acutivena* [3], flavonoids are found in the leaves of *Litsea coreana*, *Litsea japonica* and *Litsea glaucescens* [4–8], sesquiterpenes are found in the leaves and twigs of *Litsea verticillate* and *Litsea lancilimba* [9,10]. Essential oils are present abundantly in the leaves, fruits, flowers, and bark of LM and in the fruits of *Litsea glutinosa* [11]. However, the pharmacological properties of many these plants have not been thoroughly investigated by science, although these species have a lot of potential for the potential discovery of bioactive herbs and chemicals.

*Litsea* genus pant, *Litsea monopetala* is distributed abundantly throughout tropical and subtropical Asia, including Bangladesh (Fig 1) [12]. It is an evergreen tree growing from 5–20 metres tall. Leaves of this species are broadly oval or obovate to ovate-oblong, alternate and densely hairy like branchlets [12,13]. Seeds of *LM* plant is good source of oil (around 30%) and this oil possesses industrial applications (e.g., cosmetic preparations, skin conditioners and others) [11,14–17]. The bark of this species is mildly astringent, and powdered bark and roots are used externally against bruises and pains [1,2]. External application of leaf, bark, and root powder is used to treat bruising and aches [18]. Phytochemical screening of the methanolic and ethanolic extract of *LM* leaves extract revealed the presence of alkaloids, flavonoids, polyphenols, glycosides, protein, terpenoids, saponins, and carbohydrates [2,5,19–22]. Therefore, *LM* leaf extracts' pharmacological and nutraceutical potential can be better understood by estimating their total phenolic, flavonoid, and antioxidant potential. Moreover, *LM* leaf extracts can be screened chemically and pharmacologically to yield supporting information that allows them to be transformed into nutraceuticals by adding nutritional and/or health claims.

There have been reports of analgesic, antibacterial, antioxidant, antidiabetic, anti-diarrheal, anti-fungal, anti-arrhythmic and cytotoxic properties *LM* leaves extract

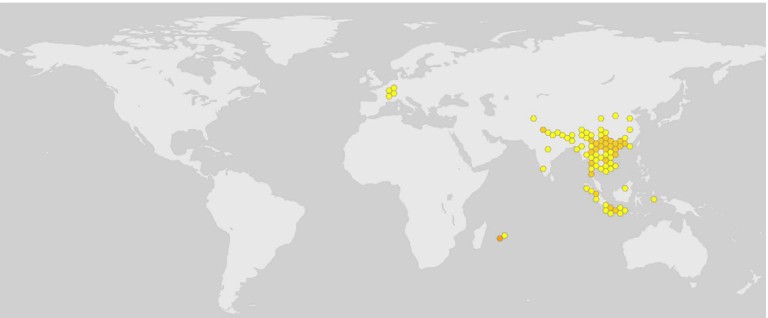

**Fig 1. Distribution of *litsae monopetala* plant in the planet.** Classical yellow and orange hexagonal symbol indicates the distribution of *LM* on the world map. Data curated from the Global Biodiversity Information Facility (https://www.gbif.org/) for illustrative purposes only.

[23–25]. Nasrin F., *et al* (2015) evaluated the antidiarrheal activity of ethanolic extracts from *LM* leaves and observed 60% inhibition of defecation at 400 mg/kg body weight dose [26]. *In vivo* analgesic, antiemetic and anxiolytic effect of methanolic extract of *L. monopetala* leaves was also evaluated in formalin induced animal model and demonstrated a statistically significant (P < 0.05) reduction in paw licking [24]. Methanolic extracts of LM leaves were studied for antioxidant, analgesic and antidiarrheal properties [23]. This study showed antioxidant ($IC_{50}$ = 223.22 µg/ml), analgesic (68.75% writhing inhibition) and antidiarrheal properties. In an *in-vitro* study, Lamichhane G *et al.* (2023) tested selected five plants, including *LM* leaves and observed remarkable antioxidant, antibacterial, anti-adipogenic, and anti-inflammatory activities from Nepal [27]. Thus, there is a strong rationale to validate the ethnomedicinal uses of *LM* leaves extract of Bangladesh origin.

DPPH assay is a common method that measures the *in-vitro* antioxidant capacity, while two standard methods acetic acid-induced and castor oil-induced animal model are also frequent to test *in-vivo* analgesic and antidiarrheal activities, respectively [23–27]. Thrombolytic effects are typically assessed through in vitro testing with human blood by measuring the percentage of the clot lysis [23–27]. In this study, we have attempted to evaluate not only the *in-vitro* antioxidant and thrombolytic, but also *in-vivo* analgesic and antidiarrheal properties of *LM* leaves in different extracts. This research would be a valuable data addition to *the effective and safe use of LM Plant in the ethnomedicinal arena.*

## Materials and methods

### Experimental animal

For this experiment, Swiss albino mice, aged 4–5 weeks, with a mean weight of 20–25 g, were used in this study. All the animals were bought from Jahangirnagar University (JU), in Bangladesh's Animal Research Lab. They were given JU-formulated food and water and kept in a typical setting for a week at the World University of Bangladesh research facility to help them acclimate.

### Experimental animal care

For this experiment, 6–8-week-old Swiss albino mice, weighing 40–45gram on average were employed. Mice were housed under controlled environmental conditions, with a temperature maintained between 20°C and 24°C, relative humidity at 30%, and a photoperiod of 12 hours of light followed by 12 hours of darkness. The animal rooms and cages were regularly cleaned, and suitable nesting material was provided within the cages to enable the mice to regulate their microclimate effectively, in addition allow them to perform their natural nesting behaviors. They were given Jahangirnagar University, Bangladesh -formulated food ad-libitum and tap water. Mice kept in a typical setting for a week at the World University of Bangladesh research laboratory to help them acclimate. We regularly observed the Body temperature, body

weight, behavioural changes (e.g., reduced exploration), and pathological changes using imaging technology, and blood oxygen saturation to implement humane endpoints by following OECD guidelines.

## Ethical approval

This study was carried out in strict accordance with the recommendations in the Guide for the Care and Use International Centre for Diarrheal Disease Research, Bangladesh (ICDDRB) guidelines, which was followed for conducting animal experiments. The World University of Bangladesh Ethical Committee Ethics of Animal Experiments read out the study through and gave its approval (Approval no # WUB/2023/242L6). The Ethical Committee on Animal Experiments at the World University of Bangladesh, in collaboration with our research team, mandated that the experimental mice used in this study would not be reused for any subsequent experiments. Upon completion of the study, all were promptly and humanely euthanized to minimize distress, fully in accordance with ethical guidelines. We strictly adhered to humane euthanasia methods and environmentally responsible handling practices, ensuring that we met the highest standards of animal welfare and ethical responsibility.

## Method of sacrifice

The method of cervical dislocation for euthanizing mice under 200 grams was performed without anesthesia, adhering to the 2020 AVMA and IACUC guidelines. Trained personnel carried out the procedure skillfully, using just the right amount of force to ensure the cervical dislocation quickly and effectively without any pain or distress. Cervical dislocation is a common method used in research for euthanizing animals quickly and humanely. It involves the severing or displacement of the cervical spine (neck) from the brainstem, which leads to immediate loss of motor function and rapid death due to the disruption of vital nervous system connections. During cervical dislocation, the thumb and index finger were positioned on either side of the animal's neck at the base of the skull while the animal lay on a table surface. On the other hand, the base of the tail is firmly and steadily pulled to cause separation of the cervical vertebrae and spinal cord from the skull. Following the procedure, we monitored the mice for over 60 seconds until the heartbeat ceased before disposal. Trained personnel monitored the mice for signs of pain or distress throughout the process. Critical assessment of the presence or absence of pain and/or distress between control and drug treated mice behavior has been taken carefully.

## Chemicals

We purchased Catechin, ferric chloride, gallic acid, trichloroacetic acid and DPPH (1,1-diphenyl, 2-picryl hydrazyl) from Sigma Chemical Co (USA). Potassium ferricyanide was purchased from May & Backer, Dagenham, UK, and ascorbic acid was acquired from SD Fine Chem. Ltd. in India. We bought potassium ferricyanide, sodium carbonate, methanol, and Folin-Ciocalteu reagent from Merck in Germany. Lyophilized streptokinase (1,000,000 IU) was purchased from Square Pharmaceuticals Ltd., Dhaka, Bangladesh. All the chemicals used in this experiment were analytical grade.

## Collection and identification of plant sample

The fresh plant was collected for Tangail district village, Bangladesh. Freshly collected branches and leaves were taken to the Bangladesh National Herbarium in Mirpur, Dhaka for authentication. The voucher specimen was added to the Bangladesh National Herbarium and given the accession number 45413.

## Drying and grinding of plant materials

The collected leaves were cleaned under clean running water to remove any remaining dirt. The samples were first allowed to dry at room temperature under the shade for a week, then dried at 50–60°C in a mechanical dryer to achieve

complete drying. The dried leaves were mechanically ground into a coarse powder. The powdered sample was stored in a sealed airtight container in a cool, dry, and dark place till further use.

## Extract preparation

Four hundred gram (400 g) of leaf powder was weighed and dissolved in 1500 ml methanol, and extracted using a Soxhlet extraction apparatus for 72 h. The methanol solvent was evaporated from the extraction by using a rotary evaporator under reduced pressure to obtain methanol crude extract. A greenish-black gooey and sticky concentrated methanol crude extract was used for phytochemical screening and pharmacological activity evaluation. The extract was stored at 40ºC in a Pharmaceutical standard refrigerator until further use. Six grams of crude extract dissolved and extracted with aqueous methanol (MSF), petroleum ether (PSF), Ethyl acetate (ESF), carbon tetrachloride (CTF), and chloroform (CSF). The fractionated amounts were PSF(1.5g), ESF(1.5g), CTF(1.4g), CSF(0.5g), and residual MSF(0.11g), respectively. All crude extracts were filtered separately through Whatman No. 41 filter paper to remove particles. The particle-free crude extract was evaporated completely by using rotary evaporator under reduced pressure to obtain dry crude extracts. The residue left in the separatory funnel was re-extracted twice following the same procedure and filtered. The combined extracts were concentrated and dried by using a rotary evaporator under reduced pressure.

## Phytochemical screenings

All crude extracts were subjected to confirmatory qualitative phytochemical screening with a slight modification of Sofowara's standard protocol for phytochemicals in the crude extracts [28]. The main classes of compounds obtained were various organic soluble fractions, including alkaloids, flavonoids, steroids, phenols, resins, glycosides, and saponins, as well.

## Estimation of total phenolic content (TPC)

By utilizing the Folin-Ciocalteu [29] reagent, the TPC was determined. Two milligrams of crude extract were liquefied with 2 ml of distilled water. In the test tube, 2 ml of diluted Folin-Ciocalteu (previously diluted 10-fold with deionized water) was mixed with 0.5 ml of extract (1 mg/ml). The mixture was left to stand at $22 \pm 2$ ºC for five minutes. Then 2.5 milliliters of 7.5% $Na_2CO_3$ was added to the test tube. To allow for color development, the mixture was gently mixed for 20 minutes without allowing any significant jerking. The UV-Vis spectrophotometer (Model: UV-1700 series) method was used to measure the color change's intensity at 760 nm. The compound's TPC was represented in the absorbance value. The final concentration of 0.1 mg/ml was used to assess the extract and standard sample. The GA equivalent, or GAE (Standard curve equation: $y = 0.0085x + 0.1125$, $R^2 = 0.9985$), was used to express TPCs as mg of GA/g of dry extract (S1 Fig).

## Determination of total flavonoid content (TFC)

We followed the aluminum chloride colorimetric method, the total flavonoid concentration of the crude extracts was determined. In brief, 0.1 ml of 10% $AlCl_3$ was added with 1.5 ml of crude extract, and then 0.1 ml of 1 M Na-acetate was subsequently added to the reaction mixture. The mixture was left to stand for half an hour. After that, 1 ml of a 1 mol/l NaOH solution was added, and double-distilled water was used to bring the mixture's final volume to 5 ml. After the mixture was let to stand for 15 minutes, the absorbance at 415 nm was determined. The calibration curve was used to determine the total flavonoid content, which was then reported as mg of quercetin equivalent per g of dry weight. Total flavonoid content was calculated from the following calibration curve:

$$Y = 0.01 X + 0.0409, R^2 = 0.9921$$

where Y is the absorbance of crude extract and X is the catechin equivalent (S1 Fig).

### Acute toxicity test

To test acute toxicity, the methanol crude extract was administered to mice at doses 200, 400, 800, 1600, and 3200mg/kg PO. We used six mice in each group. Before administration to mice, animals were kept fast for 16h. All animals were allowed free access to food and water followed by observed for 48h for symptoms of acute toxicity. The number of deaths within this period was recorded.

### Total antioxidant capacity assay

A known approach was used to calculate the total antioxidant capacity [10]. Two milliliters of phosphomolybdenum reagent solution and 1.8 milliliters of distilled water are combined with 0.2 milliliters of various plant extracts (100, 80, 600, 40, and 20 µg/ml). The mixture was incubated at 95°C for 90 minutes. After the mixture has cooled to room temperature, a reagent blank is used to detect the absorbance at 695nm. The test was performed in triplicate. The following theoretical formula was used to determine the antioxidant capacity, which is represented as ascorbic acid equivalent (AAE):

$$A = CxV/M$$

Where, A = Total content of antioxidant compounds, (mg/g) leaf extract, in AAE, C = The concentration of ascorbic acid established from the calibration curve, (mg/ml), V = The volume of extract (ml), and M = The weight of crude leaf extract (g).

### Free radical scavenging assay

Free radical scavenging activities (antioxidant capacity) of the extracts were estimated by following previously published protocol by Lamichhane, G. et al. (2023) [23]. The ability of DPPH to scavenge free radicals was demonstrated by the spectrophotometrically magnified change in colour from purple to yellow in methanol. 3.0ml of DPPH methanol solution (20 µg/ml) was combined with 2.0mL of MSF and its fractions of extracts at varying concentrations. By using a UV spectrophotometer to compare the extract's antioxidant capability to ascorbic acid (AA), the extract was shown to be capable of bleaching a purple-colored methanol solution containing DPPH radicals. Positive control was employed with AA. The mother solution with a concentration of 1000 µg/ml was created by dissolving a calculated quantity of AA in methanol. The mother solution was serially diluted to get differential concentrations ranging from 20.0 to 100 µg/ml. For extracts, mother solution (1000 µg/ml) was obtained by dissolving the measured quantity of crude extracts in methanol. Differential concentration was obtained by dilution of the mother solution in steps. Twenty microgram per milliliter of DPPH solution was obtained by weighing and dissolving 20mg of DPPH powder in methanol. The prepared solution was stored in the light-proof box together with the amber reagent bottle. Three milliliter (3.0ml) of DPPH methanol solution was combined with 2.0mL of a methanol solution of the sample (extractives/control) at varying concentrations. Using methanol as a reference, the absorbance was measured at 517nm using a UV spectrophotometer following a 30-minute reaction at room temperature in a dark environment. The following formula was used to get the percent inhibition of free radical DPPH inhibition:

$$(I \%) = (1 - \text{Absorbance of Sample/Absorbance of control}) \text{ X } 100$$

Similar procedure was repeated for the standard ascorbic acid instead of sample solution (plant extract) to obtain the percentage inhibition. The graph showing the proportion of inhibition against extract concentration was used to determine the extract concentration that would provide 50% inhibition ($IC_{50}$).

### Analgesic activity assay

To test the analgesic activity, experimental animals are randomly divided in 3 groups (n = 6). Each group received different treatment: the extract, the standard (diclofenac sodium), and the control. Diclofenac (standard) and test samples (plant

extract) are administered orally using a feeding needle. All the experimental animals received an intraperitoneal injection of 0.7% acetic acid solution (15 ml/kg), a writhing-inducing chemical, to guarantee adequate absorption of the drugs provided after a 30-minute interval. Acetic acid activates the nerve and is used to simulate writhing by releasing endogenous chemicals that cause algia. Following a 30-minute injection (IP) of acetic acid, their body contractions are observed. Then the test group received crude extract at a dose of 500 mg/kg BW and the standard group received diclofenac at a dose of 25 mg/kg BW, while the control received a diluted tween-80 solution. After five minutes, the writhing number (squirms) was meticulously counted for 15 minutes. All test samples are adjusted by diluting with distilled water to 5.0 ml volume.

## Antidiarrheal activity assay

To assess the antidiarrheal activity of the methanolic crude extract, we adopted a previously published method by Shoba and Thomas et al. (2001) [30]. Experimental animals are grouped, each consisting of six mice. The group under control: received vehicle (1% Tween 80 in water) orally at a dose of 10 ml/kg body weight; the positive control group received loperamide orally at a dosage of 3 mg/kg body weight. Test group 1 received an oral dosage of 200 mg/kg body weight of methanolic extract; test group 2 received an oral dose of 400 mg/kg body weight of methanolic extract. Oral administration of 0.5 ml castor oil was used to produce diarrhoea in each mouse 30 mins following the aforementioned treatment. Every mouse was housed in a separate cage, and white blotting paper was used to line the floor. The experiment was carried out for four hours, changing the blotting sheets every hour. The total amount of diarrheal feces (watery, unformed stool) that occurred throughout the observation period was noted. After an hour, the old absorbent papers were swapped out for new ones.

The control group's total amount of diarrheal feces was considered to be 100%. The defecation inhibition percentage was calculated using the following formula.

$$\text{\% Inhibition of defection} = \left[(A\text{-}B)/A\right] \times 100$$

A= Mean number of defecations by castor oil; B = Mean number of defecations by extract/ loperamide

## Castor oil-induced enteropooling assay

Following the Robert et al. (1976) method, we assess the intraluminal fluid buildup [31]. Experimental animals are randomly divided into four groups and six animals in each group. The Group 1 and 2 received prostaglandins (200 micrograms/kg; ip) and distilled water (Oral), respectively. The group 3 and 4 received crude extracts orally at doses of 300 and 600 mg/kg, respectively, one hour before the castor oil was administered orally. The mice were put to death and given ether anaesthesia two hours later. The small intestine was removed and weighed after the margins were secured with thread. Squeezing the intestinal material into a graduated tube allowed for the measurement of its volume. The difference between the full and empty intestines was computed after the intestine was reweighed.

## *In-vitro* thrombolytic activity test

A method published by Prasad et al.(2006) was followed to assess the thrombolytic activity of the crude extract [32]. The protocol was approved by the ethical committee of the World University of Bangladesh, Dhaka, Bangladesh. Lyophilized Streptokinase powder was purchased from Beacon Pharmaceuticals Ltd, Dhaka, Bangladesh. Five milliliters of sterile distilled water was added to the vial to reconstitute the streptokinase and mixed thoroughly. The reconstituted sample was stored as per the labeled instructions. Venous blood was drawn from the healthy human volunteers ($n = 10$) and transferred (0.5 ml/tube) to the previously weighed sterile microcentrifuge tube to form the clot. Before withdrawing the blood sample, a written agreement was taken from the volunteers. The samples are incubated at 37 °C for 45 min. After the clot formation, serum was completely removed without disturbing the clot and each tube having clot was again weighed

to determine the clot weight. A volume of 100 µl of methanol extract (10 mg/ml) was added to each microcentrifuge tube containing pre-weighed clot. As a positive control, we added 100 µl (30000 IU) of streptokinase, and as a negative control 100 µl of distilled water to the control tube. All the tubes were then incubated for 90 minutes at 37 °C and observed the clot lysis. After incubation, the fluid released was removed and tubes were again weighed to observe the difference in weight of clot lysis. The difference obtained in weight taken before and after clot lysis was expressed as a percentage of clot lysis. All the experiments are performed in at least three replicates.

$$\frac{\text{Weight of the clot after 90 minutes}}{\text{Initial weight of the clot}} \times 100 = \% \text{ of clot lysis.}$$

### Statistical analysis

Results were expressed as mean ± SEM. To determine the statistical significance one way ANOVA followed by Dunnett's multiple comparisons was performed.

## Results and discussion

### Identification and confirmation of crude phytochemicals present in the *LM* extract

Several phytochemicals such as tannins, phenols, steroids, flavonoids, alkaloids, and saponins were confirmed in the extract through qualitative tests (Table 1). Among the tested phytochemicals our results reveal tannins as the major phytocompound present in the crude extract as determined by the lead acetate test. There are reports that tannins are an interesting class of polyphenols and exert several pharmacological effects, including anti-inflammatory, antimicrobial, antioxidant, anticancer, neuroprotective, cardioprotective and anti-metabolic diseases[33]. Qualitative tests also demonstrate a moderate presence of carbohydrates, alkaloids, and flavonoids in the extract as determined by the molisch's test, mayer's test, and shinoda's test, respectively. While reducing sugars, steroids, and phenolic content were present in insignificant quantities. Interestingly, saponins was absent in the extract. However, the amount of phytochemicals in the plant extracts also depends on changes in the climate, geography, and altitude. These elements influence the amount of persistent secondary metabolites are during the adaption process. Because of these factors, the phytochemicals our study reports might differ slightly from the information found in the literature on the chosen plant[7,16,19,34].

### Estimation of total phenolic contents (TPC) in crude extract

Using gallic acid (GAE) as the standard, we observed a straight-line graph of the TPC in the crude methanol extract (Fig 2A). In particular, we tested the TPC in MSF, PSF, CTF, CSF, and ESF of leaves of *LM* crude methanol extract. The average TPC found were 165.02, 65.15, 219.18, 185.73, and 301.67 mg/g of extract, respectively (Fig 2A, Table 2). We observed a

**Table 1. Phytochemical screening of methanolic extract of LM leaves.**

| Phytochemicals | Test Method | Remarks | Description |
|---|---|---|---|
| Carbohydrates | Molisch's Test | ++ | Moderately present |
| Reducing sugars | Fehling solution test | + | Mildly present |
| Tannins | Lead acetate test | +++ | Largely present |
| Alkaloids | Mayer's test | ++ | Moderately present |
| Flavonoids | Shinoda test | ++ | Moderately present |
| Saponins | Frothing test | – | Not present |
| Steroids | Libermnn-Burchards test | + | Mildly present |
| Phenol | Ferric chloride test | + | Mildly present |

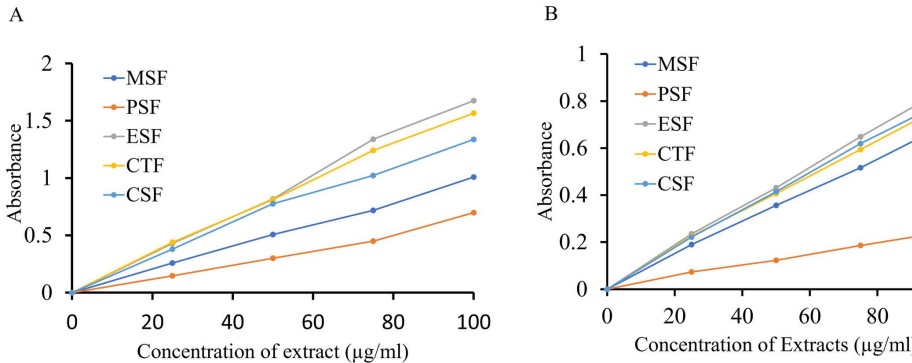

**Fig 2. Estimation of (A) Total phenolic contents; (B) Total flavonoid contents among various fractions of *LM* leaves.**

**Table 2. Total phenolic, total flavonoid contents of crude extract of LM.**

| Sample | TPC (mg/ml) Mean±SEM | TFC (mg/ml) Mean±SEM |
|---|---|---|
| MSF | 165.02±0.57 | 686.40±0.52 |
| PSF | 65.15±0.55 | 165.02±0.45 |
| ESF | 301.67±0.47 | 635.41±0.54 |
| CTS | 219.18±0.37 | 699.30±0.61 |
| CSF | 185.73±0.15 | 702.40±0.35 |

lower quantity of TPC in PSF (65.15±0.27 mg/g) while in ESF (301.67±0.40 mg/g) TPC obtained was maximum, followed by CTF and CSF. In comparison to MSF and PSF, the phenolic contents of ESF were significantly higher (p<0.05). This finding implies that all of the soluble fractions of *LM* leaves offer phenol content at varied degrees. The results of this study show that *LM* leaves might have a good source of phenolic compound with possible health advantages. Some studies showed the presence of phenolic compound and antioxidant activities of methanolic extract of *LM* leaves [19,22,27,35].

## Estimation of total flavonoid content (TFC)

Flavonoids are well known for their antioxidant properties, thus attracting considerable interest in medicinal plant search research. We used the complexometric technique to quantify total flavonoid concentration in the extract (Fig 2B). Like TPC, we observed a straight-line (Y = 0.01 X + 0.0409, R2 ≥ 0.9921) graph for the TFC concentration in the extract. We used catechin (with a concentration range of 10.0 µg/ml to 160.0 µg/ml) as the standard, and the flavonoid contents were evaluated in terms of catechin equivalent (CE) (Fig 3). The TFC was measured in the presence of other organic solvents, including PSF, CTF, CSF, and ESF. The average TFC found were 686.30, 165.01, 635.10, 699.30, and 702.40 mg/g of dry extract, respectively (Table 2). The maximum amount of TFC was found in ESF (702.30±0.35 mg/g) and the lowest amount in PSF (165.35±0.45 mg/g), followed by CSF, CTF and MSF. When compared to MSF and PSF, the flavonoid contents of ESF were considerably higher (p<0.05). The study's confirmation of noteworthy flavonoid concentrations in all soluble fractionates suggests that *LM* leaves are a rich source of flavonoids, which makes them useful for medicinal purposes [1,2,5].

## Effect of *LM* crude extract on total antioxidant capacity (TAC)

We evaluated the TAC of the crude extract in comparison to the ascorbic acid (AA). A complexometric technique involving the reduction of molybdenum (V) to molybdenum (III) was used to measure the TAC (Fig 3). Based on a known

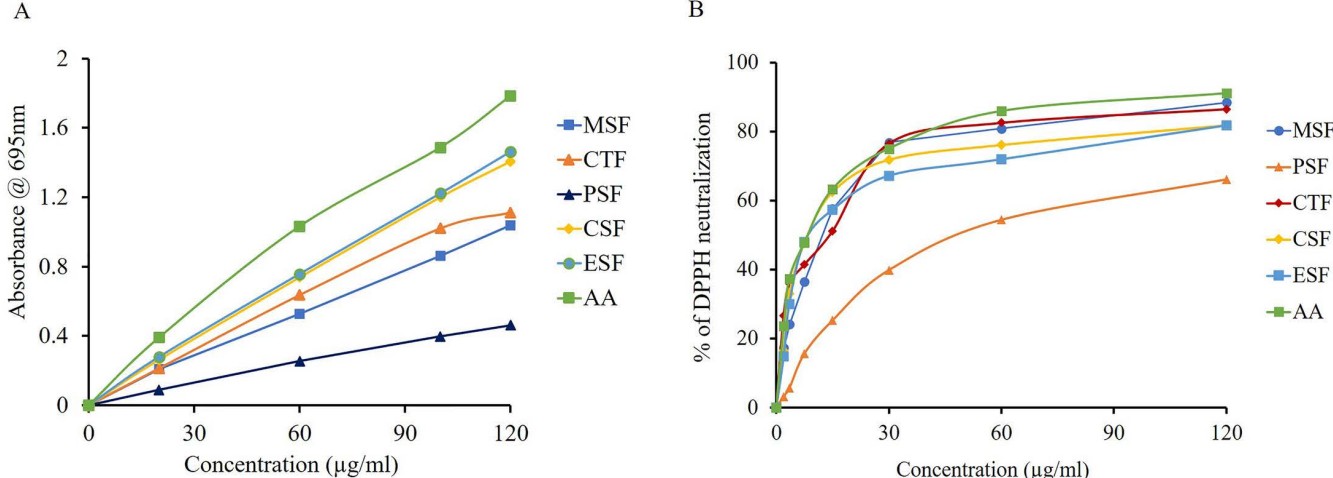

**Fig 3. In vitro analysis of antioxidant activity among various fractions of leaves of *litsae monopetala*.** (A) Total antioxidant capacity analysis; (B) DPPH free radical scavenging activity analysis. Ascorbic acid (AA) was used as the standard antioxidant. CSF, Chloroform fraction; CTF, Carbon tetra-chloride fraction; ESF, Ethyl acetate fraction; MSF, Crude methanolic extract; PSF, Petroleum ether fraction.

**Table 3. Total antioxidant capacity and DPPH free radical neutralizing property (IC$_{50}$) of crude extract.**

| Extract/ Standard | TAC (mg AAE*/g of extract) | IC$_{50}$ value (µg/ml) |
|---|---|---|
| | Mean ± SD | Mean ± SD |
| MSF | 329.78 ± 0.61[b] | 12.99 ± 0.21[c] |
| PSF | 132.53 ± 0.51[a] | 47.05 ± 0.46[d] |
| CTF | 318.04 ± 0.71[b] | 6.08 ± 0.21[a] |
| CSF | 399.19 ± 0.15[c] | 8.99 ± 0.21[b] |
| ESF | 400.90 ± 0.59[c] | 6.88 ± 0.411[a] |
| AA | 410.90 ± 0.39[c] | 8.99 ± 0.42[b] |

*Ascorbic acid equivalent; the letters a, b, c, d in the superscript indicates the level of significance, as determined by Duncan's Multiple Range Test (DMRT

concentration of AA, a straight-line graph was obtained (Fig 3). The TAC found in MSF, PSF, CTF, CSF and ESF were 329.61 ± 0.19, 132.51 ± 0.51, 318.04 ± 0.71, 399.76 ± 0.19 and 400.90 ± 0.59 mg/g of dry extract, respectively (Table 3). Except PSF, CSF, CTF, MSF and ESF showed the greater TAC (>400.90 mg/g of dry extract) that were almost close to AA. In both ESF and CSF, TAC was considerably greater (p < 0.05). The TPC and TFC are connected to the TAC of each fraction. TPC and TFC exhibited a significant connection with total antioxidant capacity (r² = 0.946, 0.965; p < 0.05) according to pearson correlation analysis. By donating a proton, plant polyphenols and flavonoids balance singlet or triplet oxygen and may shield the body from oxidative stress.

### Effect of *LM* crude extract on DPPH radical scavenging

We tested DPPH radical scavenging capacity of the crude extracts as the measure of the TAC (Fig 3). The data obtained of DPPH free radical neutralizing property (IC$_{50,}$ %) for ascorbic acid (AA), MSF, PSF, CTF, CSF, and ESF produce a straight-line graph. For MSF, PSF, CTF, CSF, and ESF IC$_{50}$ values were 12.99 ± 0.21, 47.05 ± 0.46, 6.08 ± 0.21, 8.99 ± 0.21, and 6.98 ± 0.41 µg/ml, respectively (Table 3). When compared to MSF, PSF, CSF, and AA, the IC$_{50}$ values

for ESF and CTF were significantly lower (p<0.05). The highest potential to scavenge free radicals caused by DPPH were CTF and ESF (IC$_{50}$ values 6.08 µg/ml and 6.98 µg/ml, respectively), whereas AA's IC$_{50}$ was 8.99±0.42 µg/ml. A strong association between TPC and TFC and DPPH free radical scavenging was found through correlation analysis (r$^2$=0.9331, 0.976; p<0.05). Published reports also describe the antioxidant properties of aqueous alcoholic extractives of LM's tubers, seeds, leaves, and bark. Rich amounts of flavonoids and polyphenols in LM leaves also function as proton donors and have a strong association with the plant's ability to scavenge free radical's DPPH (Fig 3) [7,16,19,22,23,34,36,37].

### Effect of *LM* crude extract on analgesic activity

An Acetic acid-induced writhing model was used to assess the analgesic activity of the crude extract. Acetic acid induces writhing upon the release of endogenous chemicals that excite the pain nerve and create analgesia. We used Diclofenac as a standard drug and compared the analgesia with our extract. The writhing response caused by acetic acid is significantly reduced by the LM extract at a dosage of 500 mg/kg body weight. In mice, leaves of LM induced 67.05% writhing inhibition, compared to 74.15% with conventional medicine Diclofenac at a dosage of 25 mg/kg body weight of the experimental animals (Table 4). When compared to control mice, the methanolic extract of LM demonstrated a notable analgesic effect. However, further research is needed to identify the original active ingredient that gives LM's methanolic extracts their analgesic effects.

### Effect of *LM* crude extract on antidiarrheal activity

We used Castor oil-induced animal models to conduct in vivo antidiarrheal experiments. Administration of crude extract to tested animals exhibited a lesser frequency of feces compared to the control group. Compared to the conventional loperamide (3 mg/kg BW), the group extract showed a lower frequency of feces (Table 5). Defecation inhibition was seen in 71.1% of the extract group compared to the control group. During a 4-hour testing period, all of the extracts demonstrated antidiarrheal efficacy as evidenced by a decrease in the feces frequency in comparison to the control group. In addition to calculating the percentage of defecation inhibition, the total number of defecations throughout 4 hours was noted. The extract showed 51.09% (200 mg/kg) and 60.0% (400 mg/kg) of suppression of defecation in a dose-dependent manner. Every extract showed statistically significant inhibition of defecation frequency and volume (*p<0.05* and *p<0.01*) (Table 6). Previous studies on phytochemicals found that the presence of tannins, alkaloids, flavonoids, saponins, sterol, and

Table 4. Evaluation of the Analgesic activity using the writhing test (n=6).

| Group | Writhing (Total) | Writhing (Mean±SEM) | Decrease compared to control (%) |
|---|---|---|---|
| Control (Tween 80) | 162 | 32.5±1.85 | – |
| Diclofenac (25mg/kg) | 37 | 7.1±1.05 | 74.25 |
| Extract (500mg/kg) | 49 | 9.6±1.01 | 67.05 |

Table 5. Effect of methanolic extract in antidiarrhoeal activity determined by feces frequency.

| Groups | Dose | Feces Frequency at 4 hours (Mean±SEM) | Inhibition (%) |
|---|---|---|---|
| Control | 10 ml/kg | 13.1±1.1 | – |
| Positive control(loperamide) | 3 ml/kg | 2.8±0.4 | 71.1% |
| Extract of LM (Test group I) | 200 ml/kg | 6.0±0.6 | 51.09% |
| Leaf extract of LM (Test group II) | 400 ml/kg | 5.0±0.5 | 60.0% |

triterpenes in medicinal plants was linked to their antidiarrheal properties [23,26]. Tannins cause the intestinal mucosa's proteins to become denatured by producing protein tanuates, which increase the mucosa's resistance to chemical modification and reduce secretion [38]. According to reports, flavonoids prevent the release of prostaglandins and autocoids, which may prevent castor oil-induced secretion and motility [31,39,40]. The study's experimental plants are abundant in tannins, alkaloids, phenols, flavonoids, steroids, and maybe additional phytochemicals. Therefore, the presence of these beneficial phytochemicals may be the cause of the plants' strong antidiarrheal effect.

We also tested antidiarrheal activity using an enteropooling assay (Table 6). Oral treatment of the extract an hour before to castor oil administration substantially (P<0.05) reduced the enteropooling at 1.18 ml (300 mg/kg) and 1.28 ml (600 mg/kg). Similar to the 1.01 ml intestinal fluid volume seen in the normal group (Table 5). After receiving castor oil therapy, the weight of the intestinal content likewise dramatically increased (in normal rats, from 1.00 g to 25.9.). Nonetheless, the weight of the intestinal content decreased somewhat as a result of the extract.

### Effect of crude extract on thrombolytic activity

We assessed the thrombolytic potential of the crude methanolic and another partitioning of LM. In contrast, a negative control condition of sterile distilled water investigated a small fraction of clot breaks down (3.24%). The following fractions showed a higher percentage of clot lysis in MSF (45.58%), PSF (40.00%), CTF (38.99%), and ESF (14.55%). Hence MSF exhibited a higher percentage of clot lysis (45.58%) in compare to standard streptokinase (69.52%).The interpretation of the data suggests that the LM extract had good thrombolytic activity; (p<0.01; p<0.05), as shown in Table 7 and Fig 4. One of the main vascular diseases that causes many heart conditions, particularly cardiovascular ischemic events, is thrombosis[41]. The purpose of the current investigation was to determine whether *LM* leaf extracts had any potential for thrombus breaking. We compared the MSF and its Kupchan fractions at a concentration of 2 mg/ml with the negative control using the positive control's results. According to our research, test sample thrombolytic activities had positive

**Table 6. Effect of methanolic extract in antidiarrhoeal activity as determined by enteropooling assay.**

| Group | Volume of intestinal content (ml) (n=6) | | Weight of intestinal content (g) (n=6) | |
|---|---|---|---|---|
| | Mean+SEM | *p-value* | Mean ±SEM | *p-value* |
| Normal saline (2ml) | 1.01±0.15 | – | 1.00±0.06 | – |
| Castor oil (2ml) | 2.99±0.18 | – | 2.59±0.5 | – |
| Extract (300mg/kg) + Castor oil (2ml) | 1.18+0.19 | *0.05* | 2.05+0.16 | *0.03* |
| Extract (600mg/kg) + Castor oil (2ml) | 1.28±0.18 | *0.01* | 2.01±0.12 | *0.01* |

**Table 7. Effect of LM leaves crude extract and Streptokinase on thrombolytic activity after 90 minutes.**

| Group | Clot lysis (%) (Mean±SEM) |
|---|---|
| Negative Control (Water) | 4.24±0.90 |
| PSF | 13.54±0.63 |
| MSF | 40.79±0.44 |
| CTF | 39.81±0.32 |
| CSF | 46.58±0.21 |
| Streptokinase (SK) | 69.52±0.11 |

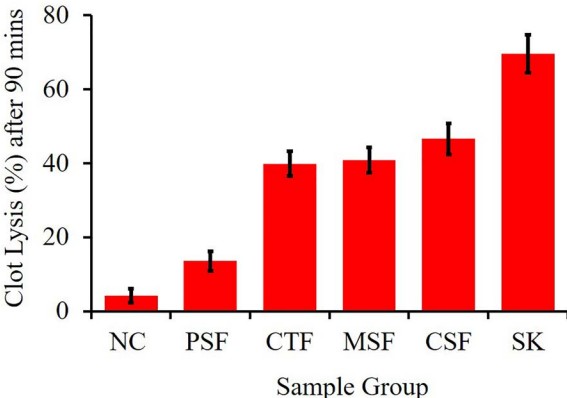

**Fig 4. Thrombolytic activities of different fractionates of _LM_ leaves.**

cascades when compared to positive and negative controls. The results suggested that _LM_ had phytochemicals that are in charge of the activity related to clot lysis.

## Conclusion

Our current study's results allow us to draw the conclusion that the _LM_ leaf extracts shown strong analgesic and anti-diarrheal properties in addition to modest antioxidant and thrombolytic activity. The study's findings indicate that more research may be conducted to identify or separate novel compounds from _LM_ leaf extracts to advance the medicinal industry.

## Supporting information

**S1 Fig. Standard curve of Gallic acid, Catechin and Ascorbic Acid.**
(TIF)

## Author contributions

**Conceptualization:** Sarder Arifuzzaman, Zubair Khalid Labu, Samira Karim.

**Data curation:** Zubair Khalid Labu, Banani Das Ani, Samira Karim.

**Formal analysis:** Sarder Arifuzzaman, Zubair Khalid Labu, Samira Karim.

**Investigation:** Sarder Arifuzzaman, Zubair Khalid Labu, Samira Karim.

**Methodology:** Sarder Arifuzzaman, Zubair Khalid Labu, Samira Karim.

**Project administration:** Md. Tarekur Rahman.

**Resources:** Zubair Khalid Labu, Banani Das Ani, Samira Karim.

**Software:** Sarder Arifuzzaman, Banani Das Ani.

**Supervision:** Sarder Arifuzzaman, Zubair Khalid Labu.

**Validation:** Sarder Arifuzzaman, Zubair Khalid Labu, Banani Das Ani.

**Visualization:** Md. Tarekur Rahman.

**Writing – original draft:** Sarder Arifuzzaman, Zubair Khalid Labu.

**Writing – review & editing:** Sarder Arifuzzaman, Zubair Khalid Labu, Md. Tarekur Rahman.

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
