## [Decision Letter · Decision Letter 0]

22 Jan 2025

PONE-D-24-49091Methanol Crude Extract of Litsea Monopetala Leaves combats oxidative stress, clot formation, inflammation and stool frequency in Animal ModelPLOS ONE

Dear Dr. Arifuzzaman,

Thank you for submitting your manuscript to PLOS ONE. After careful consideration, we feel that it has merit but does not fully meet PLOS ONE’s publication criteria as it currently stands. Therefore, we invite you to submit a revised version of the manuscript that addresses the points raised during the review process.

We look forward to receiving your revised manuscript.

Kind regards,

Waqas Khan Kayani, PhD

Academic Editor

PLOS ONE

Comments from PLOS Editorial Office: We note that one or more reviewers has recommended that you cite specific previously published works. As always, we recommend that you please review and evaluate the requested works to determine whether they are relevant and should be cited. It is not a requirement to cite these works. We appreciate your attention to this request.

1. Please ensure that your manuscript meets PLOS ONE's style requirements, including those for file naming. The PLOS ONE style templates can be found at https://journals.plos.org/plosone/s/file?id=wjVg/PLOSOne_formatting_sample_main_body.pdf and https://journals.plos.org/plosone/s/file?id=ba62/PLOSOne_formatting_sample_title_authors_affiliations.pdf.

2. Please match your authorship list in your manuscript file and in the system.

4. To comply with PLOS ONE submissions requirements, in your Methods section, please provide additional information regarding the experiments involving animals and ensure you have included details on (1) methods of sacrifice, (2) methods of anesthesia and/or analgesia, and (3) efforts to alleviate suffering.

5. We note that your Data Availability Statement is currently as follows: [All relevant data are within the manuscript and its Supporting Information files.] Please confirm at this time whether or not your submission contains all raw data required to replicate the results of your study. Authors must share the “minimal data set” for their submission. PLOS defines the minimal data set to consist of the data required to replicate all study findings reported in the article, as well as related metadata and methods (https://journals.plos.org/plosone/s/data-availability#loc-minimal-data-set-definition). For example, authors should submit the following data:- The values behind the means, standard deviations and other measures reported;- The values used to build graphs;- The points extracted from images for analysis. Authors do not need to submit their entire data set if only a portion of the data was used in the reported study. If your submission does not contain these data, please either upload them as Supporting Information files or deposit them to a stable, public repository and provide us with the relevant URLs, DOIs, or accession numbers. For a list of recommended repositories, please see https://journals.plos.org/plosone/s/recommended-repositories. If there are ethical or legal restrictions on sharing a de-identified data set, please explain them in detail (e.g., data contain potentially sensitive information, data are owned by a third-party organization, etc.) and who has imposed them (e.g., an ethics committee). Please also provide contact information for a data access committee, ethics committee, or other institutional body to which data requests may be sent. If data are owned by a third party, please indicate how others may request data access.

Reviewers' comments:

Reviewer's Responses to Questions

**Comments to the Author**

1. Is the manuscript technically sound, and do the data support the conclusions?

Reviewer #1: Yes

Reviewer #2: Partly

2. Has the statistical analysis been performed appropriately and rigorously? 

Reviewer #1: Yes

Reviewer #2: No

3. Have the authors made all data underlying the findings in their manuscript fully available?

Reviewer #1: No

Reviewer #2: Yes

4. Is the manuscript presented in an intelligible fashion and written in standard English?

Reviewer #1: No

Reviewer #2: No

5. Review Comments to the Author

Reviewer #1: To acess the antidiarrheal activity ethanolic extracts are reported in the results while the objective was to study crude methanolic extracts. This discripency needs to be clarified

Format of the references should be checed

References mentioned in the text should also be included in the bibliography

Reviewer #2: The manuscript titled “Methanol Crude Extract of Litsea Monopetala Leaves combats oxidative stress, clot formation, inflammation, and stool frequency in Animal Model” presents interesting findings on the therapeutic potential of Litsea monopetala. However, there are several points that need to be addressed to improve the clarity, structure, and accuracy of the manuscript. Below are detailed suggestions for revision:

1. Revised Sentence on Antioxidant Assessment: “To assess the antioxidant… phytochemical screening was performed to estimate the bioactive compounds (e.g.- phenol, flavonoid, alkaloids, tannins, and others) present.” This sentence should be revised to enhance clarity and precision.

2. The abstract lacks a clear structure separating the introduction, methods, results, and conclusions. Please revise the abstract to follow the typical structure of scientific abstracts: introduction, material and methods, results and conclusions.

3. To date, no study conducted in in-vivo experiments to find possible therapeutic benefits for common health problems...” This statement is inaccurate as there are multiple published studies on the in vivo effects of Litsea monopetala. The authors should review the following studies and revise their statement accordingly. Please cite these references and modify the statement to accurately reflect the existing literature.

a. In vivo antidiarrheal study of ethanolic extracts of Mikania cordata and Litsea monopetala leaves DOI: 10.3329/bjp.v10i3.23402

b. Evaluation of in vivo analgesic, antiemetic and anxiolytic effect of methanolic extract of Litsea monopetala in animal model. DOI:10.15562/phytomedicine.2019.102

c. The Analgesic Potential of Litsea Species: A Systematic Review. doi: 10.3390/molecules29092079

d. Evaluation of Antioxidant, Analgesic and Antidiarrheal Activities of Methanolic Extract of Litsea monopetala (roxb.) Leaves DOI: 10.4172/2167-065X.1000185

4. Figures 1, 3, and 4 are not clear due to low resolution. Authors need to improve the resolution of these figures to ensure they are of sufficient quality for publication. Higher-resolution images will allow for clearer interpretation of the data.

5. Figure 2: Move the standard curves for Gallic Acid, Catechin, and Ascorbic Acid to the supplementary data section. This will streamline the main manuscript while still providing essential data for reference.

6. Authors should ensure that the unit of expression for results is consistent throughout the manuscript. Please decide whether to use "µg/mL" or "µg/ml" and apply it uniformly across all sections, tables, and figures.

7. Table 1: The remarks represented by "+" and "–" signs should be clarified. Include a detailed explanation of what each symbol represents in the table's footnotes to improve

8. A comprehensive revision of the manuscript is necessary to improve the grammar, sentence structure, and overall readability. Consider using a professional language editing service or revising the manuscript thoroughly to ensure it meets the publication standards

6. PLOS authors have the option to publish the peer review history of their article (what does this mean? ). If published, this will include your full peer review and any attached files.

**Do you want your identity to be public for this peer review?** For information about this choice, including consent withdrawal, please see our Privacy Policy .

Reviewer #1: No

Reviewer #2: No

---

## [Author Response · Author response to Decision Letter 1]

1 Mar 2025

Response to academic editor and reviewer’s comments

We have addressed all issues indicated in the review report, and believed that the revised version can meet the journal publication requirements. The followings are our amendments to the reviewer’s comments written by point-by-point. We hope our present form of amended manuscript can meet with your standard as well as PLOS ONE.

Comment from Editor

1. Please ensure that your manuscript meets PLOS ONE's style requirements, including those for file naming. The PLOS ONE style templates can be found at https://journals.plos.org/plosone/s/file?id=wjVg/PLOSOne_formatting_sample_main_body.pdf and https://journals.plos.org/plosone/s/file?id=ba62/PLOSOne_formatting_sample_title_authors_affiliations.pdf.

Response: Thank you for your comment. We believe the revised manuscript will meet the PLOS ONE’s Publication requirement.

2. Please match your authorship list in your manuscript file and in the system.

Response: Thank you for your comment. The authorship in the manuscript file and the system has been rechecked and made synchronous.

Response: Thank you for your comment. This is not competing interests of data presented in this study (line 4457-458)

4. To comply with PLOS ONE submissions requirements, in your Methods section, please provide additional information regarding the experiments involving animals and ensure you have included details on (1) methods of sacrifice, (2) methods of anesthesia and/or analgesia, and (3) efforts to alleviate suffering.

Response: Thank you for your comment. Information regarding (1) methods of sacrifice, (2) methods of anesthesia and/or analgesia, and (3) efforts to alleviate suffering of the experiments animals has been discussed in detail in the materials and method section. Please have a look (line 128-142)

5. We note that your Data Availability Statement is currently as follows: [All relevant data are within the manuscript and its Supporting Information files.]

Response: Thank you for your comment. Data availability statement is followed as per PLOS ONE journal guideline. All relevant data are either within the manuscript or its supporting information files.

Comment from Reviewer #1:

To access the antidiarrheal activity ethanolic extracts are reported in the results while the objective was to study crude methanolic extracts. This discrepancy needs to be clarified

Format of the references should be checked

References mentioned in the text should also be included in the bibliography

Response:

Dear reviewer,

Thank you very much, and we appreciate your time and effort in reviewing our manuscript to improve. We addressed all the points and revised our manuscript accordingly. We are requesting you reconsider the manuscript.

We are expressing apology for the unintentional typographic mistakes and we corrected it in the revised manuscript. Anomalies in the reference format have been revised as per the journal's guidelines. The bibliography has also been revised as per references in-text citation. Please have a look (line 463-600)

Comment from Reviewer #2:

The manuscript titled “Methanol Crude Extract of Litsea Monopetala Leaves combats oxidative stress, clot formation, inflammation, and stool frequency in Animal Model” presents interesting findings on the therapeutic potential of Litsea monopetala. However, there are several points that need to be addressed to improve the clarity, structure, and accuracy of the manuscript. Below are detailed suggestions for revision:

Response:

Dear reviewer,

Thank you very much and we appreciate your time in reviewing the manuscript. We addressed all of your suggestions and revised our manuscript accordingly.

1. Revised Sentence on Antioxidant Assessment: “To assess the antioxidant… phytochemical screening was performed to estimate the bioactive compounds (e.g.- phenol, flavonoid, alkaloids, tannins, and others) present.” This sentence should be revised to enhance clarity and precision.

Response: Thank you for your comment. The sentence has been revised. Please have a look (line 28-34)

The revised texts are follows:

We assessed the antioxidant activity using DPPH (2,2-diphenyl-1-picrylhydrazyl) free radical scavenging and total phenolic content tests, while thrombolytic activity was evaluated via clot lysis assays. The in-vivo analgesic and antidiarrheal activities were tested by two standard methods e.g., acetic acid-induced and castor oil-induced animal model, respectively. Prior to in vivo and in vitro evaluation of the pharmacological activities phytochemical screening was performed to estimate the bioactive compounds (e.g., phenol, carbohydrates, reducing sugars, tannins, alkaloids, flavonoids, saponins and steroids) present.

2. The abstract lacks a clear structure separating the introduction, methods, results, and conclusions. Please revise the abstract to follow the typical structure of scientific abstracts: introduction, material and methods, results and conclusions.

Response: Thank you for your comment. The abstract has been revised as per your comment. Please have a look in the revised manuscript (line 23- 47)

3. To date, no study conducted in in-vivo experiments to find possible therapeutic benefits for common health problems...” This statement is inaccurate as there are multiple published studies on the in vivo effects of Litsea monopetala. The authors should review the following studies and revise their statement accordingly. Please cite these references and modify the statement to accurately reflect the existing literature.

Response: Thank you for your comment. We corrected the statement para in the introduction section as per your comment. Please have a look in the revised manuscript (line 87-93.)

The revised texts are follows:

There have been reports of analgesic, antibacterial, antioxidant, antidiabetic, anti-diarrheal, anti-fungal, anti-arrhythmic and cytotoxic properties LM leaves extract [23-25]. Nasrin F., et al (2015) evaluated the antidiarrheal activity of ethanolic extracts from LM leaves and observed 60% inhibition of defecation at 400 mg/kg body weight dose [26]. In vivo analgesic, antiemetic and anxiolytic effect of methanolic extract of L. monopetala leaves was also evaluated in formalin induced animal model and demonstrated a statistically significant (P<0.05) reduction in paw licking [24]. Methanolic extracts of LM leaves were studied for antioxidant, analgesic and antidiarrheal properties [23]. This study showed antioxidant (IC50 =223.22 μg/ml), analgesic (68.75% writhing inhibition) and antidiarrheal properties. In an in-vitro study, Lamichhane G et al. (2023) tested selected five plants, including LM leaves and observed remarkable antioxidant, antibacterial, anti-adipogenic, and anti-inflammatory activities from Nepal [27]. Thus, there is a strong rationale to validate the ethnomedicinal uses of LM leaves extract of Bangladesh origin.

DPPH assay is a common method that measures the in-vitro antioxidant capacity, while two standard methods acetic acid-induced and castor oil-induced animal model are also frequent to test in-vivo analgesic and antidiarrheal activities, respectively [23-27]. Thrombolytic effects are typically assessed through in vitro testing with human blood by measuring the percentage of the clot lysis [23-27]. In this study, we have attempted to evaluate not only the in-vitro antioxidant and thrombolytic, but also in-vivo analgesic and antidiarrheal properties of LM leaves in different extracts. This research would be a valuable data addition to the effective and safe use of LM Plant in the ethnomedicinal arena.

a. In vivo antidiarrheal study of ethanolic extracts of Mikania cordata and Litsea monopetala leaves DOI: 10.3329/bjp.v10i3.23402

We revised and added the reference. Please have a look in the revised manuscript (ref no 26, line 540-541)

b. Evaluation of in vivo analgesic, antiemetic and anxiolytic effect of methanolic extract of Litsea monopetala in animal model. DOI:10.15562/phytomedicine.2019.102

We revised and added the reference. Please have a look in the revised manuscript (ref no 24, line 534-536)

c. The Analgesic Potential of Litsea Species: A Systematic Review. doi: 10.3390/molecules29092079

We revised and added the reference. Please have a look in the revised manuscript (ref no 18, line 511-513)

d. Evaluation of Antioxidant, Analgesic and Antidiarrheal Activities of Methanolic Extract of Litsea monopetala (roxb.) Leaves DOI: 10.4172/2167-065X.1000185

We revised and added the reference. Please have a look in the revised manuscript (ref no 23, line 531-533)

4. Figures 1, 3, and 4 are not clear due to low resolution. Authors need to improve the resolution of these figures to ensure they are of sufficient quality for publication. Higher-resolution images will allow for clearer interpretation of the data.

Response: Thanks a lot. Figures 1, 3, and 4 have been revised and resubmitted through the system

5. Figure 2: Move the standard curves for Gallic Acid, Catechin, and Ascorbic Acid to the supplementary data section. This will streamline the main manuscript while still providing essential data for reference.

Response: Thanks for your suggestion. We moved figure 2 in the supplementary data section (S1_Fig.) and resubmitted through the system.

6. Authors should ensure that the unit of expression for results is consistent throughout the manuscript. Please decide whether to use "µg/mL" or "µg/ml" and apply it uniformly across all sections, tables, and figures.

Response: Thank you very much for pointing out this mistake. As per your comment, we revised our manuscript with “µg/ml” to improve infirmity in the manuscript. Please have a look in the revised manuscript (Fig 2, Fig 3, line 37, 221, 226-228)

7. Table 1: The remarks represented by "+" and "–" signs should be clarified. Include a detailed explanation of what each symbol represents in the table's footnotes to improve

Response: Thanks for the comment. The manuscript has been revised and remarks has been clarified with footnotes. Please have a look in the revised manuscript. (line no 632)

8. A comprehensive revision of the manuscript is necessary to improve the grammar, sentence structure, and overall readability. Consider using a professional language editing service or revising the manuscript thoroughly to ensure it meets the publication standards

Response: Thanks for the comment. The manuscript has been revised and the language has been checked. Please have a look in the revised manuscript

Sincerely

Sarder Arifuzzaman, M.S.

Sr. Lecturer

Department of Pharmacy

World University of Bangladesh

Avenue 6 and Lake Drive Road, Sector#17, Uttara, Dhaka-1230, Bangladesh

C.P.: +8801776883100

Email: sarder.arifuzzaman@pharmacy.wub.edu.bd

arifpharmju@gmail.com

---

## [Decision Letter · Decision Letter 1]

29 Apr 2025

Methanolic Crude Extract of Litsea Monopetala Leaves combats oxidative stress, clot formation, inflammation and stool frequency in Animal Model

PONE-D-24-49091R1

Dear Dr. Sarder Arifuzzaman,

We’re pleased to inform you that your manuscript has been judged scientifically suitable for publication and will be formally accepted for publication once it meets all outstanding technical requirements.

Kind regards,

Waqas Khan Kayani, PhD

Academic Editor

PLOS ONE

Additional Editor Comments (optional):

Reviewers' comments:

Reviewer's Responses to Questions

**Comments to the Author**

1. If the authors have adequately addressed your comments raised in a previous round of review and you feel that this manuscript is now acceptable for publication, you may indicate that here to bypass the “Comments to the Author” section, enter your conflict of interest statement in the “Confidential to Editor” section, and submit your "Accept" recommendation.

Reviewer #3: All comments have been addressed

2. Is the manuscript technically sound, and do the data support the conclusions?

Reviewer #3: Yes

3. Has the statistical analysis been performed appropriately and rigorously? 

Reviewer #3: Yes

4. Have the authors made all data underlying the findings in their manuscript fully available?

Reviewer #3: Yes

5. Is the manuscript presented in an intelligible fashion and written in standard English?

Reviewer #3: Yes

6. Review Comments to the Author

Reviewer #3: The authors have diligently addressed all the concerns raised by the reviewer. I recommend accepting the manuscript for publication.

7. PLOS authors have the option to publish the peer review history of their article (what does this mean? ). If published, this will include your full peer review and any attached files.

**Do you want your identity to be public for this peer review?** For information about this choice, including consent withdrawal, please see our Privacy Policy .

Reviewer #3: **Yes: ** Samreen Saleem

---

## [Editor Report · Acceptance letter]

PONE-D-24-49091R1

PLOS ONE

Dear Dr. Arifuzzaman,

I'm pleased to inform you that your manuscript has been deemed suitable for publication in PLOS ONE. Congratulations! Your manuscript is now being handed over to our production team.

Kind regards,

on behalf of

Dr. Waqas Khan Kayani

Academic Editor

PLOS ONE